# The cost-effectiveness of common strategies for the prevention of transmission of SARS-CoV-2 in universities

Zafar Zafari[1]*, Lee Goldman[2], Katia Kovrizhkin[3], Peter Alexander Muennig[3]

**1** University of Maryland School of Pharmacy, Baltimore, MD, United States of America, **2** Vagelos College of Physicians and Surgeons, Columbia University, New York City, NY, United States of America, **3** Mailman School of Public Health, Columbia University New York City, NY, United States of America

* zzafari@rx.umaryland.edu

## Abstract

### Background

Most universities that re-open in the United States (US) for in-person instruction have implemented the Centers for Disease Prevention and Control (CDC) guidelines. The value of additional interventions to prevent the transmission of SARS-CoV-2 is unclear. We calculated the cost-effectiveness and cases averted of each intervention in combination with implementing the CDC guidelines.

### Methods

We built a decision-analytic model to examine the cost-effectiveness of interventions to re-open universities. The interventions included implementing the CDC guidelines alone and in combination with 1) a symptom-checking mobile application, 2) university-provided standardized, high filtration masks, 3) thermal cameras for temperature screening, 4) one-time entry ('gateway') polymerase chain reaction (PCR) testing, and 5) weekly PCR testing. We also modeled a package of interventions ('package intervention') that combines the CDC guidelines with using the symptom-checking mobile application, standardized masks, gateway PCR testing, and weekly PCR testing. The direct and indirect costs were calculated in 2020 US dollars. We also provided an online interface that allows the user to change model parameters.

### Results

All interventions averted cases of COVID-19. When the prevalence of actively infectious cases reached 0.1%, providing standardized, high filtration masks saved money and improved health relative to implementing the CDC guidelines alone and in combination with using the symptom-checking mobile application, thermal cameras, and gateway testing. Compared with standardized masks, weekly PCR testing cost $9.27 million (95% Credible Interval [CrI]: cost-saving-$77.36 million)/QALY gained. Compared with weekly PCR testing, the 'package' intervention cost $137,877 (95% CrI: $3,108-$19.11 million)/QALY

**Data Availability Statement:** All relevant data are within the paper and its Supporting Information files.

**Funding:** Our study is funded by Columbia University Mailman School of Public Health. The funder had no roles in the design of the study and collection, analysis, interpretation of data, and writing of the manuscript.

**Competing interests:** The authors have declared that no competing interests exist.

gained. At both a prevalence of 1% and 2%, the 'package' intervention saved money and improved health compared to all the other interventions.

## Conclusions

All interventions were effective at averting infection from COVID-19. However, when the prevalence of actively infectious cases in the community was low, only standardized, high filtration masks clearly provided value.

## Introduction

In September of 2020, roughly half of United States (US) universities and colleges allowed at least some students back for in-person instruction [1–4]. Re-opening protocols for universities in the US are based on guidelines set by the Centers for Disease Control and Prevention (CDC) [5]. These include social distancing, masks, an emphasis on handwashing, and enhanced cleaning procedures in all parts of the university [5, 6]. Many universities attempted to supplement the core CDC guidelines with additional preventive interventions.

To address uncertainties surrounding the cost and effectiveness of interventions to prevent the spread of COVID-19, we developed the Columbia Covid-19 Model [7]. This is a user-accessible model that allows different universities to alter input parameters via an online interface based on their unique characteristics. Our aim was to calculate the cost-effectiveness of commonly used interventions for re-opening universities relative to implementing the CDC guidelines alone. We examined the cost-effectiveness of implementing the CDC guidelines in combination with: 1) a symptom-checking mobile phone application, 2) providing standardized, high filtration masks, 3) using thermal cameras for temperature screening at university entrances, 4) gateway polymerase chain reaction (PCR) testing, 5) and weekly PCR testing. We also modeled a 'package' intervention that combines CDC guidelines with providing a symptom-checking mobile phone application, standardized high filtration masks, and gateway PCR testing at the beginning of the semester followed by weekly PCR testing thereafter. We also developed an accompanying online tool that can evaluate novel interventions while also allowing local university decisionmakers to change the model parameters so that they more closely align with those of their own university setting [1, 8].

## Materials and methods

### Overview

The Columbia Covid-19 model is a decision-analytic model that deploys a Monte Carlo simulation. In this model, a cohort of students and a cohort of staff/faculty cycle daily through a 90-day semester [7]. As each day passes, the model calculates the risk of an infection, hospitalization, or death among the students and university affiliates.

For the present analysis, we used Columbia University as a case study because we had information on the socio-demographic characteristics of university affiliates (students, faculty, and staff who returned to campus in the Fall of 2020), novel data, and detailed cost information. The data that we collected include information from the extended contact tracing team, procurement costs, and expert input from the Public Health Committee. In addition, our team administered theory-grounded standard gamble exercises to graduate public health students at Columbia University to obtain data on risk-taking proclivities and willingness-to-pay data for

tuition when classes are held online only versus in-person (S1 Table in S1 Appendix). These students were chosen because they had studied the risks associated with contracting COVID-19 among student-aged populations. Our model allows for stepwise cost-effectiveness comparisons across interventions [9].

## Interventions

We compared the CDC recommendations alone ("status quo") with the CDC guidelines coupled together with each of the interventions under study [1]. Our online model allows the user to compare any given intervention against either the CDC guidelines or to no guidelines in place at all [7].

Our interventions were chosen because they were the most commonly used preventive modalities at the start of the Fall of 2020 semester in American universities [4]. This determination was made using a survey of universities [4] and the Columbia University Public Health Committee (a group of leading experts in infectious disease and university administrators).

The interventions fell into two categories: 1) reducing the number of potentially infectious affiliates on campus screening, and 2) reducing transmission on campus (S1 File in S1 Appendix).

**Interventions for removing potentially infectious affiliates.** *Symptom-checking mobile application*. We evaluated a requirement that university affiliates self-report COVID-19-associated symptoms using the university-mandated mobile phone application, which is available on iOS and Android systems and is required for entry to campus [10]. The symptom-checking application was designed to increase the proportion of exposed affiliates who self-isolate when they develop symptoms of COVID-19 (**Table 1**). After users attest to having no symptoms related to COVID-19, the application presents a green screen that can be shown to security guards.

*Thermal camera*. We also assessed thermal monitoring cameras at facility entry points to prevent entry of people with a fever as they enter the campus. The objective was to reduce the number of affiliates with any febrile illness, including COVID-19. Those who screen positive are subsequently screened with a tympanic membrane thermometer to reduce the number of false positive screens (see S2 Table in S1 Appendix for more information on how we modeled the intervention effect). This intervention also carries the benefit of removing affiliates who may have infectious diseases other than COVID-19.

*Gateway and weekly PCR testing*. Finally, we assessed one-time entry ("gateway") testing for SARS-CoV-2 for all affiliates with or without weekly testing for acute infection using PCR tests from a commercial provider (Broad Institute, Cambridge MA). Those who tested positive were required to quarantine in a campus facility for 14 days.

**Interventions for reducing transmission.** *Regular, disposable face mask and frequent hand hygiene*. For the status quo arm, we modeled the effects of CDC-recommended baseline measures. These included wearing face masks and frequent hand hygiene. For the effects of these interventions in preventing transmission of SARS-CoV-2, we used evidence from recent published studies, including a systematic review and meta-analysis [11, 12].

*Standardized, high filtration mask*. We assessed a policy that universities provide standardized, high quality, high filtration masks. This policy was adopted at Columbia University because university decisionmakers felt that highly effective N95 masks would be difficult to wear during class, but that some masks made or purchased by students would be less effective. The masks that we evaluated are snug fitting and dual ply. Although no efficacy data were available, the masks were assumed to fall roughly at the mid-point of surgical masks and N95 masks (**Table 2**) [11, 13]. Providing such masks would reduce the number of students using

**Table 1. Major assumptions used in modeling the cost-effectiveness of strategies to improve infection control for COVID-19 in the university setting.**

| |
|---|
| 1. The campus would be closed and classes would be held online for the remainder of the semester if the cumulative number of incident cases among students/staff reached 500.*† |
| 2. Upon presence of a super-spreader in the party, 5 or more university affiliates participating in that party (half of the average 10 university affiliates attending the party) would be exposed.† |
| 85% of students would self-isolate when they developed symptoms of COVID-19. We assumed a 10-percentage points improvement in this parameter associated with the use of symptom-checking mobile application.† This assumption was modeled probabilistically and tested in a one-way sensitivity analysis.† |
| 4. The average infected student would have an average of 10 close contacts (<6 feet for more than 10 minutes) on campus and 2 close contacts/day off campus prior to detection. ‡ |
| 5. Viral loads did not differ by sex, age, or severity of disease.† |
| 6. All wages were valued at the median hourly wage in the U.S. [9] |
| 7. When an otherwise healthy person was misdiagnosed by a test or thermal screening, the relevant indirect cost was lost time valued at the national average wage during the quarantine time [9]. |
| 8. Fevers detected using thermal cameras would be re-checked using a second method, such as a tympanic membrane thermometer.† |
| 9. The efficacy of university-provided masks was equal to the mean efficacy of the "average" mask used by the public and an N95 mask without vents (see Table 2) [13, 34]. |
| 10. Students would not commute to or from multi-generational households with older members or have direct contact with people over the age of 60.† |
| 11. We assumed that the duration of illness is 14 days and accounted for the possibility of long-term symptoms. |
| 12. In our model, we assumed over weekends, possible reductions in the number of close contacts between students would be offset by higher chances of spending time in the community and social gatherings. Therefore, we assumed that the total number of contacts would remain the same throughout the week. This assumption was based in part on survey data we collected on student behaviors.† |

*Based upon New York State guidelines.
†Based upon expert estimations from the Columbia University Public Health Committee or outside experts.
‡Based upon student survey.

homemade thin, loosely-fitting masks which were assumed to perform similarly to surgical masks [11, 13].

We also modeled a 'package' intervention that combined implementing the CDC guidelines with using symptom-checking mobile application, university-provided standardized, high filtration masks, and one-time entry gateway PCR testing plus weekly PCR testing thereafter.

## Outcome measures

We examined: 1) the incremental cost of each intervention after accounting for medical and intangible costs (e.g., productivity losses of quarantine for diagnosed or hospitalized affiliates and perceived monetary instructional value of in-person versus online classes); 2) the incremental quality-adjusted life years (QALYs) gained [9, 14]; and 3) the incremental cost-effectiveness ratio (ICER). The ICER is computed as changes in costs divided by the changes in QALYs. A QALY, which can be conceptualized as a year of life lived in perfect health, is calculated as the product of the remaining years of life and the health-related quality of life (HRQL) score [15].

## Model specification

Students and staff/faculty were treated as two separate but interacting populations with different baseline ages, average number of close contacts, exposures, risks of illness, hospitalizations, and deaths due to COVID-19 [16]. We used data from Columbia University on the age of each

**Table 2. Total costs and probabilities used as model inputs for estimating the cost-effectiveness of strategies to improve infection control for Covid-19 in a university setting with 16,000 students and 4,500 employees on campus during a 90-day semester.**

| Parameters | Baseline | Distribution* |
|---|---|---|
| *Population* | | |
| Number of students on campus† | 16,000 | - |
| Number of staff/faculty on campus† | 4,500 | - |
| *Daily number of close contacts* | | |
| Between each student and other students on campus† | 10 | Gamma (25, 2.5) |
| Between each student and staff/faculty on campus† | 1 | Gamma (4, 4) |
| Between each student and community members† | 2 | Gamma (4, 2) |
| Between each staff/faculty and students on campus† | 4 | Gamma (16, 4) |
| Between each staff/faculty on campus† | 1 | Gamma (4, 4) |
| Between each staff/faculty and community members† | 2 | Gamma (16, 8) |
| *Time values* | | |
| Incubation time ($r_{inc}$) [26, 27] | 5 days | Triangular (3, 14, 5) |
| Infectiousness to symptoms onset ($r_s$) | 2 days | Triangular (1, 3, 2) |
| Exposure to infectiousness | 3 days | Probability distribution of $r_{inc}$ minus probability distribution of $r_s$ |
| Duration of infectiousness after symptoms onset [28, 29] | 10 days | Triangular (6, 14, 10) |
| *Probabilities and rates* | | |
| Transmission rate per close contact [30] | 0.066 | Normal (0.066, 0.005) |
| Infection hospitalization rate among students [17, 31] | 0.008 | Beta (99.192, 12299.81) |
| Infection hospitalization rate among staff/faculty [17, 31] | 0.018 | Beta (98.182, 5356.374) |
| Infection fatality rate among students [17] | 0.0002 | Beta (99.98, 499799) |
| Infections mortality rate among staff/faculty [17] | 0.0015 | Beta (99.85, 66465.82) |
| Probability of long COVID-19 [32] | 0.133 | Beta (86.567, 564.313) |
| Proportion of students' compliance with stay-home order when they notice their symptoms† | 0.85 | Triangular (0.75, 0.9, 0.85) |
| Proportion of community members' compliance with wearing masks outside of campus [33] | 0.78 | Triangular (0.72, 0.78, 0.78) |
| *Direct costs (U.S. dollars in 2020 USD)* | | |
| Hospitalization [21, 34] | $23,489 | - |
| CDC guidelines [5] | | |
| Adhering to cleaning protocol costs [35] | $318,798 | - |
| Custodial staff [35] | $979,503 | - |
| Personal protective equipment [35] | $1,386,898 | - |
| Temperature cameras (see S2 Table in S1 Appendix) | $485,000 | - |
| PCR test (per test) ‡ | $45 | - |
| *Indirect costs (U.S. dollars in 2020 USD)* | | |
| Covid-19 infection without hospitalization for symptomatic employee who either got detected or self-quarantined (losses of productivity over 2 weeks of self-isolation) | $2,800 | Gamma (100, 0.036) |
| Covid-19 hospitalization among employees (losses of productivity over 3 weeks) | $4,200 | Gamma (100, 0.024) |
| Lost tuition value per day for online vs. in-person classes among students (Calculated from a student survey average tuition for the Fall of 2020 semester at Columbia University. See S1 and S2 Tables in S1 Appendix for more details) | $46 | |
| *Intervention effects* | | |
| CDC guidelines | | |

(*Continued*)

**Table 2.** (Continued)

| Parameters | Baseline | Distribution* |
|---|---|---|
| Hand washing/sanitizer (incidence rate ratio of infection) [12] | 0.64 | - |
| Regular mask use (odds ratio of infection) [11] | 0.33 | - |
| Hand washing/sanitizer plus regular mask use (odds ratio of infection) [11] | 0.21 | Beta (78.669, 293.816) |
| Symptom checking application (percentage points change in compliance of university affiliates quarantine upon noticing symptoms for Covid-19) | 10% | Triangular (0.75, 0.9, 0.85)+0.1 |
| Standardized masks (combined effect with frequent handwashing/sanitizing, odds ratio of infection) (see S2 Table in S1 Appendix) [11, 36] | 0.128 | Beta (87.072, 593.178) |
| Test for SARS-CoV-2 | | |
| Sensitivity | 0.95 | - |
| Specificity | 1 | - |
| *Health-related quality of life* | | |
| Losses of QALYs associated with a COVID-19 symptomatic case [37] | 0.008 | Beta (99.192, 12299.81) |
| Losses of QALYs associated with a long COVID-19 infection [37] | 0.034 | Beta (96.566, 2743.61) |
| Losses of QALYs associated with a COVID-19 hospitalization [37] | 0.020 | Beta (97.970, 4776.154) |
| Losses of QALYs associated with a COVID-19 death among student population (adjusted for average age at death, age-dependent QALYs of the US general population, and discounting future values at 3%) | 23.94 | Normal (23.94, 2.40) |
| Losses of QALYs associated with a COVID-19 death among employee population (adjusted for average age at death, age-dependent QALYs of the US general population, and discounting future values at 3%) | 18.33 | Normal (18.33, 1.83) |

Note: A close contact is defined as person-to-person contact < 6 feet for > 10 minutes. See S2 Table in S1 Appendix for further details on the model inputs.

*For triangular distributions, the parameters listed are lower limit, upper limit, and mode; for normal distributions, parameters are mean and standard deviation; for beta distribution, parameters are shape 1 and shape 2; and for gamma distributions, parameters are shape and rate.

†Expert opinion based on video conferences with the Public Health Committee at Columbia University, which is comprised of a range of infectious disease experts and administrators.

‡Costs reflect actual costs paid by Columbia University including personnel.

student, staff, and faculty member. We obtained age-related risks of hospitalization and death from the published literature and from the CDC [17, 18]. We then calculated the weighted average risks separately for students and staff/faculty by multiplying their age distribution by age-related risks of hospitalization and death.

We divided the simulation cohort into five states: susceptible (those who have not developed the disease and are at risk), exposed (those who are exposed but not yet infectious or symptomatic), infected (those who are currently infected and contagious), recovered (those who were infected in the past but are currently recovered), and death. The states of disease pathways are presented graphically in **Fig 1**. The cycle length of the model was one day, and the time horizon was over the semester (90 days).

Our model computed the daily probability of becoming infected among susceptible population based on the average number of close contacts, the transmission rate per close contact,

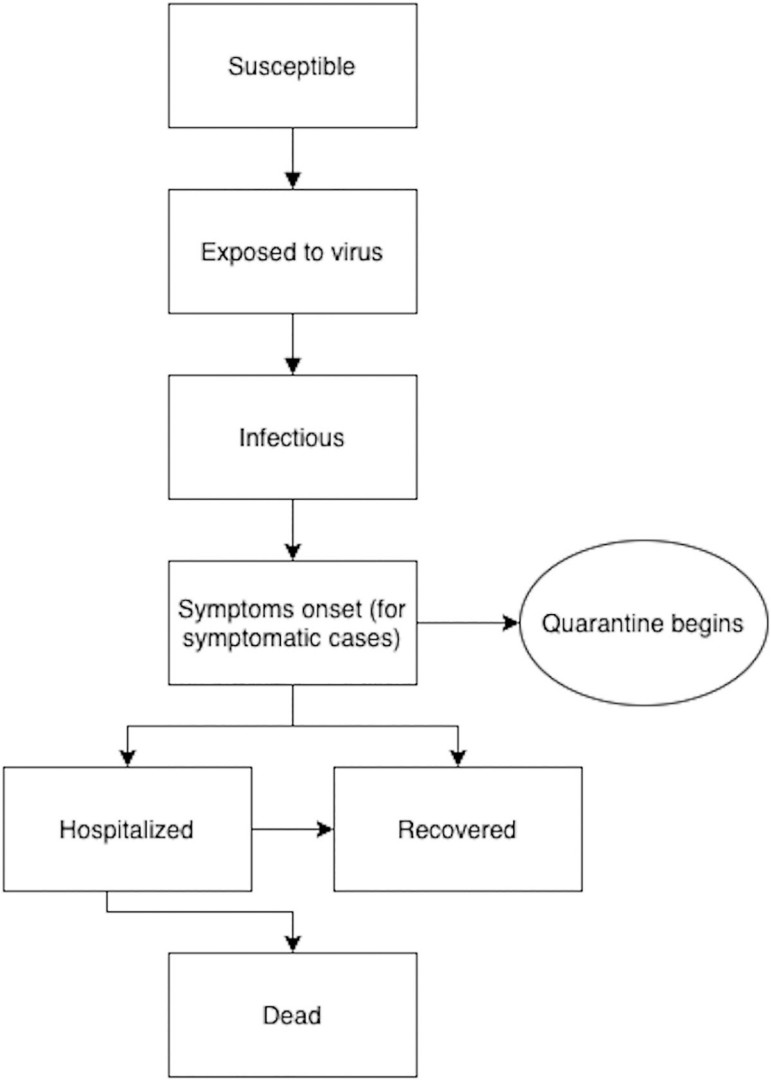

**Fig 1. Graphical representation of the states of disease pathways.**

and the estimated prevalence of infectious cases inside and outside of the campus [16]. For each susceptible student and staff/faculty, the probability of becoming infected outside of campus was calculated as follows:

$$P[\text{infection outside of campus}|\text{simulation unit } i] = (1 - (1 - p_c.r)^{c_o^i}),$$

where $p_c$ represents the prevalence of infectious COVID-19 cases in local community outside of the campus; $r$ is the transmission rate per close contact; and $c_o^i$ represents the average number of daily close contacts that each simulation unit $i$ (students or staff/faculty) makes in the local community outside of the campus.

The prevalence of actively infectious cases is an adjusted estimate of the proportion of people who, on any given day, might plausibly transmit SARS-CoV-2 to a close contact. The prevalence was calculated by dividing the number of actively infectious cases within New York City (NYC) neighborhoods within which the university is situated by the number of residents within same neighborhoods.

To obtain the prevalence of actively infectious cases, we first obtained the daily incident cases reported to the New York State Department of Health for the NYC-defined neighborhoods of interest. This number underestimates the actual incident cases on any given day because: 1) some people are asymptomatic; and 2) many people with symptoms will not be tested for COVID-19 [19, 20]. To determine the actual daily incident cases in the community, we applied a multiplier of 5, which was estimated using a COVID-19 projection model also in use by the CDC [19, 20]. We then added the incident cases of the current day to those from the past days who are still infectious. The surrounding community was defined as the area around the university in which students tend to live, in this case the official boundary for NYC-defined neighborhoods within which the university resides.

We also accounted for the proportion of population wearing face masks outside of campus. We assigned a multiplier factor, $1 - C_o . RR_{wearing\ mask}$; where $C_o$ represents the compliance rate with wearing face masks in the local community outside of the campus; and $RR_{wearing\ mask}$ represents a risk reduction associated with wearing face masks.

Similarly, the probability of becoming infected inside the campus was calculated as follows:

$$P[infection\ inside\ campus|simulation\ unit\ i] = 1 - (1 - p_s(t).r)^{c_s^i}.(1 - p_e(t).r)^{c_e^i};$$

where $p_s(t)$ and $p_e(t)$ represents the prevalence of infectious cases among students, and staff/faculty, respectively, at time $t$; $r$ is the transmission rate per close contact; and $c_s^i$ and $c_e^i$ represents respectively the average number of daily close contacts that each simulation unit $i$ makes with students and staff/faculty on campus.

We also modeled the probability of a super-spreader event based upon the prevalence of actively infectious cases in the community, the daily probability of students' participation in a party within the community, and the average number of attendees in each community party (S2 File in S1 Appendix).

Once infected, three consecutive phases of disease progression were possible, denoted as the time between: 1) the primary exposure and infectiousness; 2) infectiousness and onset of symptoms; and 3) symptom onset until the end of infectiousness (**Table 2**). For asymptomatic infected affiliates, the model excluded the second phase. At the end of the third phase, infected affiliates were classified as 'recovered.' In addition, the infected affiliates were exposed to a chance of illness, hospitalization, incurring costs, changes in HRQL, and a probability of death [21].

Lost productivity and leisure time were valued at the average American wage [22]. Intangible costs associated with online versus in-person instruction were valued using a survey administered to students who had experienced learning in each format. Risk tolerance was assessed using a standard gamble exercise (for details refer to the S1 Table in S1 Appendix). We tested the effect of the perceived value of online vs. in-person instruction in the one-way sensitivity analysis in which the value of tuition was varied from 0% to 100%.

The model accounted for interventions that: 1) remove infected affiliates from the university community (screening interventions); or 2) reduce SARS-CoV-2 transmission while on campus (e.g., wearing face masks), computed as the product of the adjusted odds ratio of infection associated with the intervention and the background odds of infection in the absence of the intervention.

The campus would close, and instruction would switch to online-only learning for the remainder of the semester, if the model reached a total of 500 cumulative cases of COVID-19 cases among students/staff/faculty over the semester.

### Analysis

We ran a probabilistic analysis using a Monte Carlo simulation with 1,000 iterations. In each iteration, all model parameters were simultaneously sampled from their probabilistic distribution. We assessed 3 scenarios of the prevalence of actively infectious cases of COVID-19: "low prevalence" (roughly 0.1%); "moderate prevalence" (1%); and "high prevalence" (2%) to represent a range of values seen across the US over the Fall semester. We calculated the stepwise cost-effectiveness comparisons, which provide information on the value of incrementally investing in strategies (e.g., investing in the most cost-effective strategy, and then adding the next most cost-effective strategy to that). We also conducted one-way sensitivity analyses to evaluate those variables that produced a large influence on the ICER. We used the common maximum willingness-to-pay threshold of $200,000 per QALY gained as a point of reference in our sensitivity analyses [9, 14, 23]. The willingness-to-pay threshold is a hypothetical reference point against which one can compare the ICER of an intervention to a maximum value society is willing to pay for one QALY gained [9, 14, 23]. In addition, we ran a series of multi-way sensitivity analyses on core model parameters including number of close contacts, transmission rate per close contact, willingness-to-pay value, compliance with wearing masks in the community, and prevalence of actively infectious cases in the community. Our model was built on the R statistical platform (The R Foundation, Inc) [7].

## Results

### Predicted number of infections

At a 0.1% prevalence of actively infectious cases in the community, 968 out of the 20,500 university affiliates in our model would contract COVID-19 over the 90-day semester if no CDC guidelines were implemented. At a prevalence of 1% and 2%, infections would rise to 4,598 and 7,865 infections, respectively. When the CDC guidelines were implemented alone, infections dropped to roughly 482 (0.1%), 3,982 (1%), and 7,430 (2%), respectively.

### Stepwise cost-effectiveness of additional interventions relative to CDC guidelines alone

**0.1% prevalence of actively infectious cases.**   At this prevalence, requiring standardized, high filtration masks in addition to implementing the CDC guidelines both saved money and resulted in a gain in QALYs compared with: 1) implementing CDC guidelines alone; or 2) implementing CDC guidelines in combination with the symptom-checking mobile application, thermal cameras, and gateway testing. Compared with standardized, high filtration masks, weekly PCR testing plus CDC guidelines produced additional costs of $10,235,673 and resulted in 1.1 QALYs gained. This produces an ICER of $9,273,023/QALY gained. Compared with weekly PCR testing, the 'package' intervention cost $40,958 and gained 0.3 QALYs for an ICER of $137,877/QALY gained. **Fig 2** shows the efficiency frontier curve for the cost-effectiveness of each of the interventions under study at a 0.1% prevalence of actively infectious cases. The stepwise calculations of ICERs from the least to the most effective intervention along with the 95% credible intervals around the outcomes' point estimates are presented in **Table 3**. The ICER values in terms of incremental cost per infection averted are presented in the S3 Table in S1 Appendix. In addition, the probabilistic results in terms of the cost-effectiveness planes and cost-effectiveness acceptability curves are presented in the Online S1 and S2 Figs in S1 Appendix.

**1% prevalence of actively infectious cases.**   At this prevalence, the 'package' intervention produced both monetary savings and QALYs gained compared with either implementing the

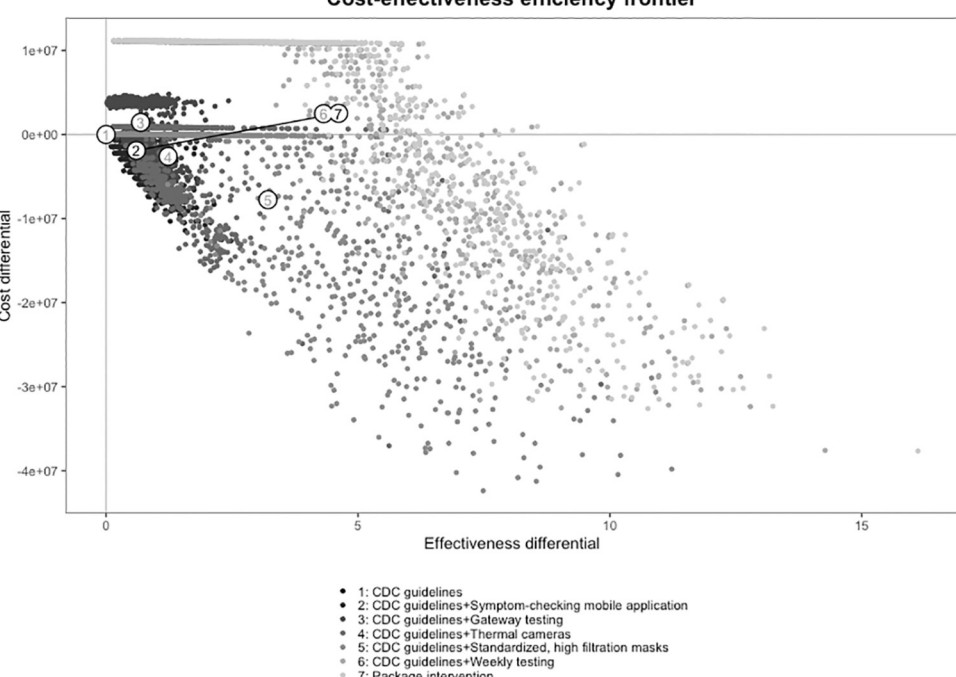

**Fig 2. Efficiency frontier curve for cost-effectiveness of strategies for the prevention of transmission of SARS-CoV-2 in universities.** The efficiency frontier curve presents the incremental cost of each intervention under study in constant 2020 US dollars relative to the change in effectiveness as measured in quality-adjusted life years (QALYs). Each intervention is paired with the Centers for Disease Control and Prevention (CDC) guidelines. Each strategy is represented by a dot in a consistent greyscale, with the CDC guidelines in black and the multi-component "package" intervention in the lightest gray. Note: CDC guidelines = the Centers for Disease Control and Prevention guidelines for preventing the transmission of COVID-19 in a university setting. The "package" intervention combines the CDC guidelines with using the symptom-checking mobile application, standardized masks, gateway PCR testing, and weekly PCR testing.

CDC guidelines alone or implementing the CDC guidelines in combination with the symptom-checking mobile application, thermal cameras, standardized masks, gateway PCR testing, and weekly PCR testing (**Table 3**).

**2% prevalence of actively infectious cases.** As with a prevalence of 1%, at a prevalence of 2%, the 'package' intervention resulted in both monetary savings and QALYs gained compared with all other interventions, including implementing the CDC guidelines alone and implementing the CDC guidelines in combination with the symptom-checking mobile application, thermal cameras, standardized masks, gateway testing, and weekly testing. (**Table 3**.)

## Sensitivity analyses

**Prevalence of actively infectious cases in the community.** Up to the point that the prevalence of actively infectious cases in the community reached 0.22%, the use of standardized, high filtration masks in addition to implementing CDC guidelines provided the highest value given the threshold of $200,000/QALY. When the prevalence exceeded 0.22%, weekly PCR testing in addition to implementing CDC guidelines and the package of interventions both provided better value for money compared with standardized masks.

**Value of online instruction.** Varying the perceived value of online vs. in-person tuition did not substantively change the model outcomes. Even when the perceived value of online-

**Table 3. Model outcomes for average number of days that the university will remain open, costs, Quality-Adjusted Life Years (QALYs), and Incremental Cost-Effectiveness Ratio (ICER).**

| | Days university open | Number of infections | Incremental costs ($) | Incremental QALYs | ICER ($/QALY) | ICER ($/QALY), without dominated |
|---|---|---|---|---|---|---|
| **100 Cases/100,000** | | | | | | |
| CDC guidelines | 79 (37, 90) | 482 (62, 1054) | Reference | Reference | Reference | |
| Symptom-checking mobile application plus CDC guidelines | 81 (40, 90) | 437 (58, 1026) | -$1901841 (-$8325490, $1835) | 0.60 (0.06, 1.55) | -$3165450 | |
| Thermal cameras plus CDC guidelines | 81 (41, 90) | 430 (56, 1016) | $3345729 ($824777, $4622951) | 0.08 (-0.21, 0.39) | $39500535 | |
| Gateway testing plus CDC guidelines | 83 (47, 90) | 388 (21, 950) | -$4043021 (-$11416977, -$1863169) | 0.55 (-0.16, 2.34) | -$7398283 | |
| **Standardized masks plus CDC guidelines** | **89 (72, 90)** | **236 (31, 696)** | **-$5154184 (-$26890309, -$851424)** | **1.98 (-0.24, 5.91)** | **-$2601899** | **-$2601899** |
| Weekly testing plus CDC guidelines | 90 (90, 90) | 152 (17, 373) | $10235673 (-$2162557, $11062938) | 1.10 (0.14, 4.89) | $9273023 | $9273023 |
| 'Package' intervention | 90 (90, 90) | 129 (16, 309) | $40958 ($2811, $82009) | 0.30 (0.00, 0.90) | $137877 | $137877 |
| **1000 Cases/100,000** | | | | | | |
| CDC guidelines | 25 (11, 90) | 3982 (459, 5929) | Reference | Reference | Reference | |
| Symptom-checking mobile application plus CDC guidelines | 26 (11, 90) | 3930 (429, 5922) | -$782349 (-$6185102, $104257) | 0.71 (0.05, 4.66) | -$1106337 | |
| Thermal cameras plus CDC guidelines | 27 (12, 90) | 3844 (416, 5810) | $916464 (-$386891, $3733634) | 1.19 (0.19, 2.32) | $772052 | |
| Standardized masks plus CDC guidelines | 38 (15, 90) | 3331 (234, 5624) | -$10329971 (-$46263518, -$2416245) | 6.85 (0.23, 37.30) | -$1508613 | |
| Gateway testing plus CDC guidelines | 40 (17, 90) | 3257 (154, 5498) | $130466 (-$6035111, $7891180) | 0.95 (-5.26, 6.76) | $137319 | |
| Weekly testing plus CDC guidelines | 52 (20, 90) | 2620 (126, 5251) | -$2928562 (-$15088412, $10351925) | 8.53 (0.27, 21.74) | -$343318 | |
| **'Package' intervention** | **57 (22, 90)** | **2377 (119, 5091)** | **-$2906565 (-$8356499, $82892)** | **3.23 (0.03, 9.64)** | **-$900687** | **-$900687** |
| **2000 Cases/100,000** | | | | | | |
| CDC guidelines | 13 (7, 29) | 7430 (4656, 9841) | Reference | Reference | Reference | |
| Symptom-checking mobile application plus CDC guidelines | 13 (7, 32) | 7412 (4531, 9838) | -$46978 (-$731014, $213804) | 0.25 (0.02, 1.24) | -$186482 | |
| Thermal cameras plus CDC guidelines | 14 (7, 34) | 7274 (4340, 9687) | $567255 (-$152899, $1086572) | 1.95 (1.30, 2.94) | $291003 | |
| Standardized masks plus CDC guidelines | 21 (8, 90) | 6861 (456, 9660) | -$6694723 (-$41433630, -$1522290) | 5.69 (-0.82, 49.96) | -$1177028 | |
| Gateway testing plus CDC guidelines | 28 (10, 90) | 6312 (299, 9432) | -$4288175 (-$30250266, $1167593) | 7.45 (0.88, 46.64) | -$575598 | |
| Weekly testing plus CDC guidelines | 34 (11, 90) | 5820 (245, 9243) | $270282 (-$15340524, $10349996) | 6.60 (0.30, 39.83) | $40941 | |
| **'Package' intervention** | **36 (12, 90)** | **5649 (231, 9148)** | **-$1321270 (-$4488178, $207549)** | **2.33 (0.06, 7.20)** | **-$567571** | **-$567571** |

All the results are the average of the 1,000 simulation runs in a probabilistic Monte Carlo simulation. ICERs were calculated as average incremental costs over average incremental QALYs in the Monte Carlo simulations and were calculated in a stepwise approach (each intervention was compared against the intervention with the next lower costs if the comparator intervention was not dominated or ruled out because of an extended dominance). Negative ICERs in this table represent a cost-saving scenario, indicating the comparator saves money and improves health compared with the baseline intervention. Under each actively infectious case prevalence scenario, the most likely cost-effective intervention at the willingness-to-pay value of $200,000/QALY was presented in bold text.

CDC: Centers for Disease Control and Prevention. CDC guidelines included social distancing, mask use, frequent handwashing, and sanitization of spaces. For the probabilistic results see S1 and S2 Figs in S1 Appendix).

*Costs and ICERs include monetary value of in-person vs. online-only instructions which were derived from a student survey at Columbia University. For details see S1 and S2 Tables in S1 Appendix.

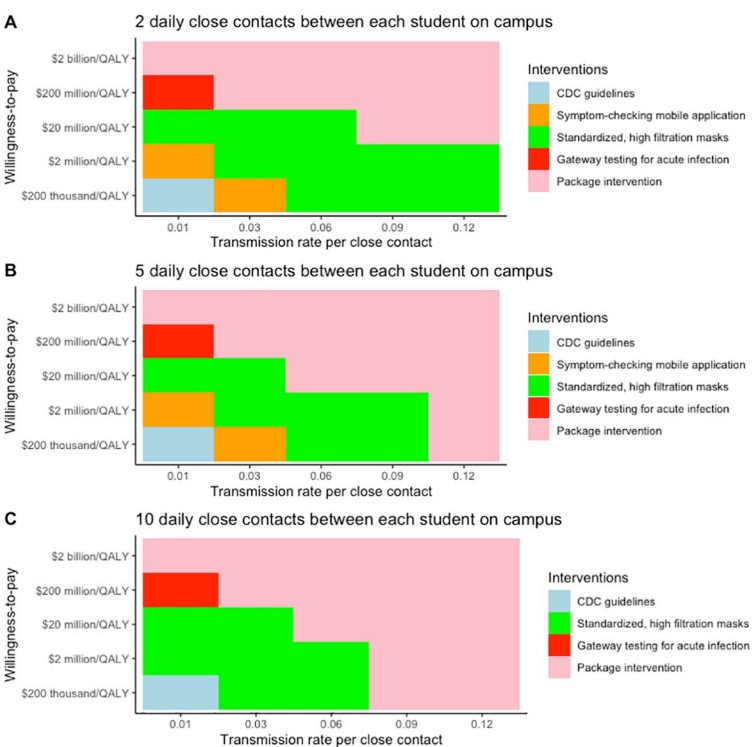

**Fig 3. Multi-way sensitivity analysis identifying the most cost-effective intervention at different values of the number of close contacts between students on campus, the transmission rate per close student contact, and willingness-to-pay at a 2% prevalence of actively infectious cases in the community.**

only tuition was equal to that of in-person classes, the use of standardized, high filtration masks in addition to implementing CDC guidelines provided the highest value.

**Influence of transmission rate, close contacts, and mask use.**   Although the prevalence of actively infectious cases was the most important driver of cost-effectiveness, the transmission rate, the number of close contacts, and the general use of masks among members of the surrounding community were important drivers of cost-effectiveness.

For example, the value of the package of interventions was sensitive both to the number of close contacts per student on campus and the transmission rate. At the base case 0.1% prevalence of actively infectious cases, increases in the transmission rate changed the most cost-effective intervention from standardized, high filtration masks to the 'package' intervention at the willingness-to-pay of $200,000/QALY. In addition, increasing the daily number of close contacts between students from 10 to 14 made the 'package' intervention the most cost-effective approach. However, standardized, high-filtration masks remained the most cost-effective intervention at a 0.1% prevalence of actively infectious cases when the average number of close contacts between students decreased from 10 to 2. **Fig 3** depicts the most cost-effective intervention at different values of the transmission rate, daily number of close contacts between students, and willingness-to-pay. Additional multi-way sensitivity analyses at a prevalence of 1% and 2% are available in the Online S3 and S4 Figs in S1 Appendix.

Reducing the proportion of people in the community who are compliant with wearing regular face masks to 25% or below changed the most cost-effective intervention to the 'package' intervention. A multi-way sensitivity analysis between the proportion of people wearing face masks in the community, the prevalence of actively infectious cases in the community, and willingness-to-pay value is presented in the Online S5 Fig in S1 Appendix.

At Columbia University, including or excluding faculty and staff over 65 years of age or 70 years of age did not have a substantial impact under any scenario because of their relatively small numbers. Finally, reducing the threshold value for the cumulative number of infections to cause campus closure (classes turning into online-only instruction upon campus closure) by 50% (from 500 to 250) made the 'package' intervention the most cost-effective approach because the 'package' intervention would keep the campus open for more days, therefore providing more monetary instructional value from in-person vs. online only classes compared with the other interventions.

## Discussion

Our model showed that the prevalence of actively infectious cases of COVID-19 in the neighborhood surrounding the university was the most important driver of cost-effectiveness when CDC guidelines were in place. At a prevalence of 0.1% (e.g., as in New York in July 2020), the most value would be realized from requiring university affiliates to use the university-provided standardized, high filtration masks in addition to implementing the CDC guidelines. When the prevalence exceeded 0.22%, the 'package' intervention provided the most value. However, variables such as the number of contacts between affiliates, the transmission rate per close contact, and face mask use in the community were also important determinants of the cost-effectiveness of the interventions we studied. As shown in **Fig 3,** reducing the number of close contacts per person and the use of face masks had a substantial influence on the likelihood of the spread of disease and therefore the cost-effectiveness of interventions to reduce the spread. Readers are encouraged to change the model inputs to suit their particular university characteristics using the online version of the model [7].

Our results are in line with a recent study by Paltiel and colleagues that recommended testing for SARS-CoV-2 when the prevalence is 0.2% [24]. We also found that for prevalence estimates of 0.22% or above, the 'package' intervention, which requires the one-time entry testing and weekly testing thereafter, would provide the highest cost-effectiveness value at Columbia University. Nevertheless, we adopted different modeling approach and assumptions surrounding: 1) infection fatality rate, 2) risk of transmission on campus, 3) the number of close contacts/student, and 4) our use of cost/QALY gained as an outcome measure rather than cost/case averted. If students have a higher number of close contacts or live in multi-generational households, we expect less cost-effective interventions to increase in value.

Our assessed infection fatality rate (0.02% for students and 0.15% for staff/faculty, **Table 2**) was smaller than the average rate for the U.S. (0.5%) [17, 24] because the population of both the students and staff/faculty was younger than the general population. Users of our online model should be careful to define risks specific to their university demographics.

Universities should consider standardizing the masks that students wear, such that their fit and filtration are superior to what students would choose to purchase on their own [11, 25]. Such standardized, high quality masks can provide the largest value, especially when the prevalence of actively infectious cases in the community is low. For example, Columbia University provided two $2 2-ply masks to each student [8].

When the prevalence of actively infectious cases in the community is high or when the average student has more close contacts, the chances of early campus closure increase. When the university is closed early, the money spent on any interventions goes to waste, and large indirect costs associated with online-only instruction are incurred. Therefore, any intervention to reduce the possibility of students attending mass events should be prioritized.

The major limitation of our analysis was the considerable uncertainty in parameter estimates. For example, estimates of infection fatality rates can quadruple when hospitals are

overwhelmed with cases [17, 20]. However, the model was generally robust to different parameter inputs and assumptions for interventions. The variables that we used as inputs to the model should be adjusted as new information and new strains of SARS-CoV-2 emerge. In addition, there were considerable uncertainties surrounding factors outside of campus, such as the enforcement of more restrictive measures when infections rose in the community. We therefore held the prevalence of actively infectious cases of COVID-19 in the community where a university is situated as a constant throughout the semester. The model will not perform well when the university population is large relative to the surrounding community.

Another limitation was that universities vary considerably with respect to socio-demographic composition and risk-taking among students. The standard gamble exercises we used were administered to students who may be more risk adverse than other students. We accounted for differences in student risk preferences by varying the number of assumed contacts between students, both on and off campus in sensitivity analyses. Finally, our model greatly underestimates risk for universities in which many students commute to and from multi-generational households.

When tailored to the conditions within which the university operates, our model should provide a robust estimate of the cost-effectiveness of interventions to prevent the spread of COVID-19. As COVID-19 becomes a seasonal illness that is complicated by variants of the virus, our model can be used by university decisionmakers to ascertain how much of an investment will be necessary to manage risk.

## Supporting information

**S1 Appendix.**
(DOCX)

## Acknowledgments

We would like to acknowledge the help and contributions of Wafaa El-Sadr, Melanie Bernitz, Steven Shea, Wan Yang, Jeffery Shamen, and the Public Health Committee for Reopening Columbia University.

## Author Contributions

**Conceptualization:** Zafar Zafari, Lee Goldman, Peter Alexander Muennig.

**Formal analysis:** Zafar Zafari.

**Funding acquisition:** Lee Goldman, Peter Alexander Muennig.

**Investigation:** Peter Alexander Muennig.

**Methodology:** Zafar Zafari, Peter Alexander Muennig.

**Resources:** Lee Goldman.

**Software:** Zafar Zafari.

**Supervision:** Peter Alexander Muennig.

**Validation:** Zafar Zafari, Katia Kovrizhkin.

**Visualization:** Zafar Zafari, Katia Kovrizhkin, Peter Alexander Muennig.

**Writing – original draft:** Zafar Zafari.

**Writing – review & editing:** Lee Goldman, Katia Kovrizhkin, Peter Alexander Muennig.

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
