## [Decision Letter · Decision Letter 0]

5 May 2021

PONE-D-21-07113

The cost-effectiveness of common strategies for the prevention of transmission of SARS-CoV-2 in universities

PLOS ONE

Dear Dr. Zafari,

Thank you for submitting your manuscript to PLOS ONE. After careful consideration, we feel that it has merit but does not fully meet PLOS ONE’s publication criteria as it currently stands. Therefore, we invite you to submit a revised version of the manuscript that addresses the points raised during the review process.

Thank you for submitting your manuscript to PLOS ONE. It considers an interesting and timely topic. However, the reviewers have identified a number of areas requiring revision and improvement. Please address these carefully.

We look forward to receiving your revised manuscript.

Kind regards,

Kevin Schwartzman

Academic Editor

PLOS ONE

Journal Requirements:

"This study is funded by Columbia University Mailman School of Public Health. The

funder had no roles in the design of the study and collection, analysis, interpretation of data, and

writing of the manuscript."

Reviewers' comments:

Reviewer's Responses to Questions

**Comments to the Author**

1. Is the manuscript technically sound, and do the data support the conclusions?

Reviewer #1: Partly

Reviewer #2: Yes

2. Has the statistical analysis been performed appropriately and rigorously? 

Reviewer #1: Yes

Reviewer #2: Yes

3. Have the authors made all data underlying the findings in their manuscript fully available?

Reviewer #1: Yes

Reviewer #2: Yes

4. Is the manuscript presented in an intelligible fashion and written in standard English?

Reviewer #1: No

Reviewer #2: Yes

5. Review Comments to the Author

Reviewer #1: Prior to my comments, I will justify my scores above.

For (1) I selected "partly" because there are some methodological questions I have. The conclusions drawn do represent the findings presented.

For (2) I selected "Yes" but I think this is with caveats. The authors have missed an opportunity to analyze multiple interventions at once and there are some questions with underlying assumptions.

For (3) I selected "Yes" as the model is open source and parameters are listed.

For (4) I selected "No" but really this is "in between" - there are very well written parts and other parts that could use proofreading for flow and grammar. Since no copyediting is provided by PLoS, I think this is the safest score.

OVERVIEW

Zafari and colleagues detail an analysis of various interventions implemented in an American university during the Fall 2020 semester across three scenarios of assumed active, infectious COVID-19 prevalence in the community. The question is certainly interesting and useful beyond the American setting as the 2021 Fall semester approaches and countries without widespread vaccination may consider returns to school. However, I have concerns. Some of the model assumptions don’t appear justified or could be flawed; I believe the analyses are incomplete and represent only simple scenarios of implementation which are unlikely to be what are done in practice (single interventions rather than multiple); and there are issues with flow and clarity and some sections of the manuscript appearing to be “artifacts” of previous iterations (e.g., estimating community prevalence has no relevance to this paper, but is mentioned in reasonable detail).

COMMENTS

Abstract

• Specify this is in the USA (“Most universities that re-opened in the USA for in-person…”)

• Second sentence appears to be a missed edit – remove “To determine”

• Methods are lacking and need to be added. What type of model used, costs considered, etc.

• Length is <200 words but max is 300. Please add more detail in all aspects of methods/results/interpretation. Are these interventions additive? Or used in a silo? It is tough to interpret any results presented in the abstract without information such as this.

Intro

• Please add the specific aim(s) of the present study.

Methods

• Typo in first sentence (“a Markov model a Monte Carlo simulation”)

• Specify what is in the novel survey data (in “Overview”)

• The final sentence of “Overview” is strange. Does this analysis/paper follow CHEERS guidelines for reporting in CEA studies? I am not sure the ideas linked in the sentence are related.

• Suggest you number the additional interventions (on top of CDC guidelines) being modeled to help readability and allow readers to more easily track what is being compared. As a reviewer… some formatting to indicate subsections would have been helpful.

• Who is the manufacturer of the SARS-CoV-2 test kits? What specimen was assumed to be used? Did you take into account other specimen types that are similarly sensitive (e.g., NP swabs, saliva, gargle)? In any event, the base analysis parameters (specimen type, manufacturer of kit, and sens/spec) should be present – were costs of specimen collection considered? These can be substantial, particularly for NP swabs (collection alone estimated to be ~$10USD with swabs, media, PPE, and nurse time). *I see in appendix a reference for sens/spec of 95% each – in fact the reference suggests specificity of 100% (https://sites.broadinstitute.org/safe-for-school/assay-performance); is this a typo? Specificity of 95% for PCR is unacceptable given low prevalence (1%), you would expect 5 false positives for every 1 true positive. Please clarify.

• It is not clear if the “one way sensitivity analyses” referred to in the “Interventions” section actually refer to interventions evaluated or sensitivity analyses?

• Regarding the formula for calculating the prevalence – I suggest there may be something amiss. I agree with multiplying the number of cases recorded each day by 5 to account for underreporting… but then multiplying that value by 7 is a bit puzzling and may lead to systematically underestimating TRUE prevalence when prevalence is declining (the ratio is prevalence is underestimated ~3x higher than the rate of decrease—e.g., detected cases dropping by 0.5% per day leads to underestimation in prevalence of 1.5%); if cases are increasing, you will overestimate prevalence in the same way. This stems from the assumption that prevalence is ONLY related to cases occurring that day – which is not true. What would be more accurate is to sum the cases from the current day and the previous 6 days to estimate prevalence. This point aside… It is not clear to me how this enters the present analysis since prevalence estimates were fixed—suggest remove unnecessary info.

• Can the authors provide a model figure of the state transitions possible and the pathways experienced? This would help supplement the text description.

• Table 2 is mentioned before Table 1.

• Provide reference for the risk tolerance exercise.

• “simultaneously changed” – do you mean all parameters were simultaneously sampled from their assigned probabilistic distribution?

• Did you run the model with the ‘baseline’ parameters and then run the model probabilistically to get the credible intervals? Or was the point estimate derived from PSA? Please specify and give the percentile from the PSA used to create the credible interval (and point estimate, if appropriate).

• “We also conducted 1-way sensitivity analyses on variables that produced a large influence on the ICER” – do you mean you ran these analyses to determine variables that produced a large influence on the ICER?

• In looking at the parameters and distributions used – limiting the daily number of close contacts via a triangular distribution is a questionable assumption. We know some people have very few close contacts… but others (a minority) have a great deal. I think this parameter would be better served with a gamma distribution to capture this “long tail” in the N contacts distribution.

• Continuing on parameters… the effect of handwashing is HUGE – larger than masks. I wonder about this assumption which appears to come from a 2008 sys review and is based on SARS studies in largely healthcare settings (perhaps not generalizable to schools/community settings)—in fact one of the studies that was done in the community found no significant effect of handwashing. There are few cases of documented fomite transmission of SARS-CoV-2, which is where handwashing would really be beneficial – in fact most ‘supposed cases of fomite transmission’ I have seen are more likely to be explained by aerosols. I suggest the authors re-consider the effect size in light of data related to SARS-CoV-2.

• Any packages in R used to help develop the model?

Results

• I would imagine what is most useful to policymakers and university personnel would be the effect of multiple interventions on COVID-19. It is hard to think of scenarios where testing is used, but other interventions are not (as these are usually progressive). The authors must consider this.

• Moreover, the authors should consider showing their findings on an efficiency frontier, which would make the claims of extended dominance easier to see and allow a more visual comparison of the various interventions.

• I think Table 3 would benefit from stating the cumulative number of covid-19 cases occurring—to allow direct comparison with what is mentioned in the first section of results. This could also be visualized on a graph/plot.

• Credible intervals are quoted, but was this analysis done in a Bayesian framework? Is this the correct terminology – it seems analysis was done in a frequentist framework to this reviewer.

• The value of online instruction sensitivity analysis is not mentioned in the methods as an explicit sensitivity analysis. It is also not entirely clear to this reviewer how to interpret the findings.

• The finding of N close contacts being a big driver of the CE of weekly testing is highly interesting.

Discussion

• The cost of masks quoted ($4) is extremely high! A 3-ply surgical mask can be procured for <$0.50 at volume. I was not able to verify this cost at the provided reference.

• Would suggest you end with a summary of findings and major take home message.

Reviewer #2: This is a detailed, excellent study of COVID19 mitigation measures on a university campus. With corrections of errors that I found and implementation of a several presentation recommendations, I feel that it meets criteria for publication.

Presentation recommendations:

* please list all 7 interventions early in the methods, and if possible in the abstract. I kept reading 7 interventions, but didn't actually find out what they were until several pages into the paper.

* the word "coupling" refers to pairing of 2 items. I suggest replacing it throughout the paper with "comparing" or deleting it. In doing incremental cost effectiveness, you are comparing a more effective strategy with a less effective one, after ordering all strategies by effectiveness.

* please bold or underline section headings. Often, they were on the next line without paragraph breaks, which made me wonder whether they were incomplete sentences.

* please provide more detail on the methods used to cost the interventions. This would be acceptable to place into an appendix, but it still needs to be in the paper.

* in Table 3, order in each prevalence section the intervention strategies by least to most effective. Since you used QALYs lost, the interventions having the greatest QALYs lost should be at the top and least at the bottom. Incremental changes in costs and QALYs are then calculated by comparing each row to the preceding one. While I determined from the write-up in the results section that you did arrive at the correct conclusions for your data, it is more apparent to readers if you present your analysis and results in the standard manner (as described above).

* since the difference in QALY is very small between strategies, I think that presenting COVID19 cases prevented would be more interesting than presenting QALYs, especially in a population less likely to die from COVID19. Most often in cost effectiveness analyses, both cases prevented and QALYs (gained) are presented as outcomes.

Here are the instances that I interpreted as errors:

* In the first paragraph of the results, you state that prevalence of 0.1% results in 351 cases. However, 351/20,500 = 1.7%. Likewise for 230/20,500 = 1.1% not 0.1%. and 1664/20,500=8.1% and 3126/20,500=15.2%.

* on p. 7, you state that you used 7 days for infectiousness. Table 2 lists exposure, symptoms, and infectiousness, with infectiousness of 9 days after symptom start. CDC lists the incubation period as 2-14 days, and infectiousness of 15 total days capturing 95% of all infectiousness. So, if 7 days was used, that is too low and doesn't match your table 2. You should consider using the standard 14 or 15 days of infectiousness.

* it appears that your base case used 2-3 contacts per student. While you did conduct a sensitivity analysis in Figure 1 on this, with up to 10 contacts per student, I would have used 10 contacts per student as the base case. In tuberculosis studies, TB cases average 15 airborne contacts per case at diagnosis. I would think that COVID19 would result in a similar number of contacts, and possibly more for gregarious students.

6. PLOS authors have the option to publish the peer review history of their article (what does this mean?). If published, this will include your full peer review and any attached files.

Reviewer #1: No

Reviewer #2: **Yes: **Suzanne Marks

---

## [Author Response · Author response to Decision Letter 0]

29 Jun 2021

Dear Dr. Schwartzman,

Thank you very much for the opportunity to respond to the helpful reviewer comments, which we feel greatly improved our manuscript. We have responded to each comment in a point-by-point response letter below. We have made extensive revisions to the manuscript. To accommodate the additional text required to respond to the reviewer comments, we have edited other sections of the document to reduce the word count.

Thank you for your time and help with our manuscript.

Best,

Peter Muennig and Zafar Zafari

 

From the editor

 [Response] Thank you. We have checked our formatting to ensure that it adheres to the journal’s requirements.

"This study is funded by Columbia University Mailman School of Public Health. The

funder had no roles in the design of the study and collection, analysis, interpretation of data, and writing of the manuscript."

 [Response] Thank you. We deleted text from the manuscript that referred to funding. As requested, we updated the funding statement in the cover letter.

Reviewer 1: 

Zafari and colleagues detail an analysis of various interventions implemented in an American university during the Fall 2020 semester across three scenarios of assumed active, infectious COVID-19 prevalence in the community. The question is certainly interesting and useful beyond the American setting as the 2021 Fall semester approaches and countries without widespread vaccination may consider returns to school. However, I have concerns. Some of the model assumptions don’t appear justified or could be flawed; I believe the analyses are incomplete and represent only simple scenarios of implementation which are unlikely to be what are done in practice (single interventions rather than multiple); and there are issues with flow and clarity and some sections of the manuscript appearing to be “artifacts” of previous iterations (e.g., estimating community prevalence has no relevance to this paper, but is mentioned in reasonable detail).

[Response] We would like to thank the reviewer for his/her time to review our paper and excellent comments. We have addressed all the comments in-depth in a point-by-point response below. We believe addressing the comments have significantly increased the quality of our paper.

As to your point regarding the model assumptions, we now addressed all your comments in the revised version including updating model parameters and their statistical distributions (e.g., updating the distribution of number of contacts to gamma). Please see our point-by-point response below for details.

For your point regarding single rather than multiple interventions, please note that each of the intervention of our model are in combination with implementing the CDC guidelines. For example, we compared weekly testing in addition to implementing the CDC guidelines with implementing CDC guidelines alone. We agree with the reviewer that in real practice some universities may opt to implement multiple interventions at once. Therefore, to accommodate the reviewer’s comment, in the revised version of the paper, we added the ‘package’ intervention, which is a combination of implementing CDC guidelines, using a symptom-checking mobile application, providing standardized, high filtration mask, gateway testing, and weekly testing. Therefore, in the revised version, we now have 7 interventions in total as follows: 1) CDC guidelines alone, 2) CDC guidelines combined with symptom-checking mobile application, 3) CDC guidelines combined with standardized, high filtration masks, 4) CDC guidelines combined with thermal cameras, 5) CDC guidelines combined with gateway testing, 6) CDC guidelines combined with weekly testing, and 7) the ‘package’ intervention.

We also build a transparent online platform for our model so that in case a university opts to implement an intervention that was not modeled in our study (or implement multiple interventions at once), the user can investigate any desired, user-defined input for effectiveness and cost of a ‘hypothetical intervention’ in our web application. 

For your point regarding the prevalence of disease in the areas surrounding the university, please note that we needed this parameter to model probability of infection when student/staff make contacts with people in the surrounding community outside of campus. The user can simply define this prevalence in the online application given their unique context. As suggested, we have now removed the redundant iterations of this throughout the paper. 

In the new version of the paper, we have carefully copyedited the text. We have also asked a professional academic writer to edit and proofread our paper. We hope our careful response to all the points raised meets the approval of the reviewer.

COMMENTS

Abstract

1. Specify this is in the USA (“Most universities that re-opened in the USA for in-person…”)

[Response] Thank you. We made this change.

2. Second sentence appears to be a missed edit – remove “To determine”

[Response] Thank you. Corrected.

3. Methods are lacking and need to be added. What type of model used, costs considered, etc.

[Response] In response to your concerns, we have expanded the Methods section, adding information on the model used and the costs that were considered.

4. Length is <200 words but max is 300. Please add more detail in all aspects of methods/results/interpretation. Are these interventions additive? Or used in a silo? It is tough to interpret any results presented in the abstract without information such as this.

[Response] We have made the suggested changes and expanded the Abstract. Please see changes throughout the abstract.

Intro

5. Please add the specific aim(s) of the present study.

[Response] Thank you. We added specific aims to the last paragraph of the introduction.

Methods

6. Typo in first sentence (“a Markov model a Monte Carlo simulation”)

[Response] Thank you. This typo has been corrected.

7. Specify what is in the novel survey data (in “Overview”)

[Response] Thank you. We have now expanded the relevant section of the paper for clarity and have provided additional survey data in the online appendix.

8. The final sentence of “Overview” is strange. Does this analysis/paper follow CHEERS guidelines for reporting in CEA studies? I am not sure the ideas linked in the sentence are related.

[Response] Thank you. We have edited that sentence.

9. Suggest you number the additional interventions (on top of CDC guidelines) being modeled to help readability and allow readers to more easily track what is being compared. As a reviewer… some formatting to indicate subsections would have been helpful.

[Response] Thank you. We have added headers and sub-headers. We also created a separate section for the interventions under study. 

10. Who is the manufacturer of the SARS-CoV-2 test kits? What specimen was assumed to be used? Did you take into account other specimen types that are similarly sensitive (e.g., NP swabs, saliva, gargle)? In any event, the base analysis parameters (specimen type, manufacturer of kit, and sens/spec) should be present – were costs of specimen collection considered? These can be substantial, particularly for NP swabs (collection alone estimated to be ~$10USD with swabs, media, PPE, and nurse time). *I see in appendix a reference for sens/spec of 95% each – in fact the reference suggests specificity of 100% (https://sites.broadinstitute.org/safe-for-school/assay-performance); is this a typo? Specificity of 95% for PCR is unacceptable given low prevalence (1%), you would expect 5 false positives for every 1 true positive. Please clarify.

[Response] We obtained the test parameters from Columbia University, including the cost, sensitivity, specificity, and manufacturer. These have changed somewhat since we originally submitted the paper for review. Based on your comment and our recent communication with the university’s Public Health Committee, we have updated this information and re-ran the model.

We updated the cost per test to $45 including collection, personnel and supplies. This is also in line with your suggested additional $10 for collection (previously was $31). Also, as per your suggestion, we updated the test’s specificity to 100%. It is not clear why the manufacturer’s claimed test specificity (95%) is different from what might be expected from a PCR test. Therefore, we changed the specificity of 100% as a model input parameter in order to put it more in line with other manufacturers and your suggestion.

11. It is not clear if the “one way sensitivity analyses” referred to in the “Interventions” section actually refer to interventions evaluated or sensitivity analyses?

[Response] At the time that we conducted the HVAC and far-UVC analyses in the old version of the paper, there was significant disagreement among infectious disease epidemiologists as to the extent to which SARS-CoV-2 was primarily transmitted by aerosols or by fomites. We therefore conducted 1-way sensitivity analyses on the percentage of infections attributed to aerosol transmission. The intent of the one-way analyses on these two preventive modalities was to show the reader whether assumptions about the proportion of all infections attributed to aerosol infections influenced the ICER. In the new version of the paper, to avoid confusion, we have removed these two interventions from the paper.

For the core model parameters, we have a section for the sensitivity analysis.

12. Regarding the formula for calculating the prevalence – I suggest there may be something amiss. I agree with multiplying the number of cases recorded each day by 5 to account for underreporting… but then multiplying that value by 7 is a bit puzzling and may lead to systematically underestimating TRUE prevalence when prevalence is declining (the ratio is prevalence is underestimated ~3x higher than the rate of decrease—e.g., detected cases dropping by 0.5% per day leads to underestimation in prevalence of 1.5%); if cases are increasing, you will overestimate prevalence in the same way. This stems from the assumption that prevalence is ONLY related to cases occurring that day – which is not true. What would be more accurate is to sum the cases from the current day and the previous 6 days to estimate prevalence. This point aside… It is not clear to me how this enters the present analysis since prevalence estimates were fixed—suggest remove unnecessary info.

[Response] This is an excellent point, and we fully agree. We exactly did what the reviewer suggests. We now recommend that the prevalence be computed using the current day plus the past days of the entire infectiousness period, as suggested by the reviewer.

Our online model allows universities to run the model based upon local prevalence. We have also edited and shortened the corresponding section avoiding any unnecessary information. 

13. Can the authors provide a model figure of the state transitions possible and the pathways experienced? This would help supplement the text description.

[Response] Thank you for the suggestion. We added Figure 1 to show the states of the disease pathways.

14. Table 2 is mentioned before Table 1.

[Response] Thank you for catching this. Corrected.

15. Provide reference for the risk tolerance exercise.

[Response] Added, thank you.

16. “simultaneously changed” – do you mean all parameters were simultaneously sampled from their assigned probabilistic distribution?

[Response] Yes. We have now modified the text as per your suggestion. 

17. Did you run the model with the ‘baseline’ parameters and then run the model probabilistically to get the credible intervals? Or was the point estimate derived from PSA? Please specify and give the percentile from the PSA used to create the credible interval (and point estimate, if appropriate).

[Response] The point estimates were derived from PSA. Based on your comment, we have now reported all the PSA results including the 95% credible intervals in table 3 as well as the cost-effectiveness planes and cost-effectiveness acceptability curves in the Online Appendix.

18. “We also conducted 1-way sensitivity analyses on variables that produced a large influence on the ICER” – do you mean you ran these analyses to determine variables that produced a large influence on the ICER?

[Response] Yes. We have modified the text accordingly.

19. In looking at the parameters and distributions used – limiting the daily number of close contacts via a triangular distribution is a questionable assumption. We know some people have very few close contacts… but others (a minority) have a great deal. I think this parameter would be better served with a gamma distribution to capture this “long tail” in the N contacts distribution.

[Response] Thank you for the suggestion. In the new analyses, we changed the distributions of number of contacts to gamma. Accordingly, we updated the results, figures, and tables.

20. Continuing on parameters… the effect of handwashing is HUGE – larger than masks. I wonder about this assumption which appears to come from a 2008 sys review and is based on SARS studies in largely healthcare settings (perhaps not generalizable to schools/community settings)—in fact one of the studies that was done in the community found no significant effect of handwashing. There are few cases of documented fomite transmission of SARS-CoV-2, which is where handwashing would really be beneficial – in fact most ‘supposed cases of fomite transmission’ I have seen are more likely to be explained by aerosols. I suggest the authors re-consider the effect size in light of data related to SARS-CoV-2.

[Response] Thank you. Please note that in the previous model, the effect of face masks was still larger than that of handwashing. Table 2 reports the odds ratios (OR) of infection rather than reduction in infection relative to no intervention. For face masks, the OR was 33% (a ~67% reduction) and the OR for handwashing was 45% (a ~55% reduction). However, we agree with this reviewer that the gap in efficacy should be larger, and that fomite transmission likely represents a minority of cases. In light of changes in our knowledge of transmission that has emerged since our original submission, we conducted a more recent search for the effect of frequent handwashing in the community settings. One recent study reported the adjusted relative incidence of infection of 0.64 (~%36% reduction) for frequent handwashing. As suggested by the reviewer, we have changed the OR of frequent handwashing. This parameter does not exert much influence on our model outcomes, so even if handwashing is ineffective at preventing the spread of COVID-19, it should not greatly change the results.

21. Any packages in R used to help develop the model?

[Response] The packages that we used from R were to generate statistical distributions. The rest of the code was written by ZZ. We make the code available on GitHub.

Results

22. I would imagine what is most useful to policymakers and university personnel would be the effect of multiple interventions on COVID-19. It is hard to think of scenarios where testing is used, but other interventions are not (as these are usually progressive). The authors must consider this.

[Response] Based on the reviewer’s request, we have now added a ‘package’ of interventions that combines implementing the CDC guidelines with using the symptom-checking mobile application, standardized, high filtration mask, gateway testing, and weekly testing. This package of interventions did appear to be more cost-effective as a whole than most of the individual interventions within the package. 

There are many possible combinations of interventions. Single interventions that are less cost-effective when compared with the CDC guidelines alone are likely to remain less cost-effective when added on the margin of the package of interventions. Therefore, we also present data for single interventions. Note also that all of the individual interventions are presented in combination with implementing the CDC guidelines.

23. Moreover, the authors should consider showing their findings on an efficiency frontier, which would make the claims of extended dominance easier to see and allow a more visual comparison of the various interventions.

[Response] Thank you. We have now added an efficiency frontier plot. Please see the new Figure 2 in the revised paper.

24. I think Table 3 would benefit from stating the cumulative number of covid-19 cases occurring—to allow direct comparison with what is mentioned in the first section of results. This could also be visualized on a graph/plot.

[Response] Thank you. As suggested, we have now included the number of infections in the table. We also report the incremental costs per infection averted in the online appendix.

25. Credible intervals are quoted, but was this analysis done in a Bayesian framework? Is this the correct terminology – it seems analysis was done in a frequentist framework to this reviewer.

[Response] You are correct that the analysis was done from a frequentist approach. However, since the 95% interval was informed from a Monte Carlo simulation, it was called credible interval.

26. The value of online instruction sensitivity analysis is not mentioned in the methods as an explicit sensitivity analysis. It is also not entirely clear to this reviewer how to interpret the findings.

[Response] We now mention this and provide additional text to guide the reader on how to interpret the findings. We now mention the SA around value of online instruction in the Methods:

“We tested the effect of the perceived value of online vs. in-person instruction in the one-way sensitivity analysis in which the value of tuition was varied from 0% to 100%.”

As well as in the results:

“Value of online instruction. Varying the perceived value of online vs. in-person tuition did not substantively change the model outcomes. Even when the perceived value of online-only tuition was equal to that of in-person classes, the use of standardized, high filtration masks in addition to implementing CDC guidelines provided the highest value.”

27. The finding of N close contacts being a big driver of the CE of weekly testing is highly interesting.

[Response] Thank you. We now highlight this in the discussion. In addition, we ran the model with additional close contacts per student, recognizing that students who live in universities near their family home may have more contacts than the average student at Columbia. This was done in response to another reviewer’s comments. Other universities (particularly those where students live at home with parents) may have higher numbers of close contacts.

Discussion

28. The cost of masks quoted ($4) is extremely high! A 3-ply surgical mask can be procured for <$0.50 at volume. I was not able to verify this cost at the provided reference.

[Response] We agree. We used the actual cost per mask that Columbia paid for standardized, high filtration masks plus the cost of their distribution (Columbia distributes masks by mail). They are more expensive than surgical masks. However, the original cost was obtained at the height of the pandemic when masks were more expensive. In response to the reviewer’s concern, we re-consulted the Columbia University Public Health Committee. The cost per mask has since fallen to $2, a figure we update in the revised model.

29. Would suggest you end with a summary of findings and major take home message.

[Response] Thank you. We have done so. We have added:

“When tailored to the conditions within which the university operates, our model should provide a robust estimate of the cost-effectiveness of interventions to prevent the spread of COVID-19. As COVID-19 becomes a seasonal illness that is complicated by variants of the virus, our model can be used by university decisionmakers to ascertain how much of an investment will be necessary to manage risk.”

Reviewer #2: This is a detailed, excellent study of COVID19 mitigation measures on a university campus. With corrections of errors that I found and implementation of a several presentation recommendations, I feel that it meets criteria for publication.

[Response] We would like to thank the reviewer for her time to review our paper and excellent comments. We have gone in-depth and addressed all the comments in a point-by-point response below. We believe addressing the comments have significantly increased the quality of our paper. 

Presentation recommendations:

1. please list all 7 interventions early in the methods, and if possible in the abstract. I kept reading 7 interventions, but didn't actually find out what they were until several pages into the paper.

[Response] Thank you. We now mention each of the interventions in the abstract and at the beginning of the Methods section. 

2. the word "coupling" refers to pairing of 2 items. I suggest replacing it throughout the paper with "comparing" or deleting it. In doing incremental cost effectiveness, you are comparing a more effective strategy with a less effective one, after ordering all strategies by effectiveness.

[Response] Thank you. As suggested, we also removed the word coupling and re-written most of the paper for clarity. We assumed that most universities would implement the CDC guidelines as a minimum, so the CDC guidelines serve as the “status quo” comparator. By “coupling,” we meant pairing the intervention with CDC guidelines relative to implementing the CDC guidelines alone. This allowed us to estimate the cost-effectiveness of each intervention on the margin of what we believe to be the status quo. In the revised version, based on a comment from another reviewer, we removed HVAC systems and far-UVC lighting and added a ‘package’ intervention that combines implementing CDC guidelines with using symptom-checking mobile application, standardized, high filtration mask, gateway testing, and weekly testing all at once.

We made changes to the paper throughout and modified the intervention section in the revised paper. 

3. please bold or underline section headings. Often, they were on the next line without paragraph breaks, which made me wonder whether they were incomplete sentences.

[Response] Thank you for your suggestion. We have now either bolded or underlined the headers and sub-headers throughout the paper to make them distinct from the rest of the text. We have also added additional headers and sub-headers in response to comments from another reviewer.

4. please provide more detail on the methods used to cost the interventions. This would be acceptable to place into an appendix, but it still needs to be in the paper.

[Response] Thank you. We have now expanded the costs section both in the main text and in the online appendix (please see Online S1 File and S2 Table). The costs were informed either from published data or from the Columbia University Public Health Committee.

5. in Table 3, order in each prevalence section the intervention strategies by least to most effective. Since you used QALYs lost, the interventions having the greatest QALYs lost should be at the top and least at the bottom. Incremental changes in costs and QALYs are then calculated by comparing each row to the preceding one. While I determined from the write-up in the results section that you did arrive at the correct conclusions for your data, it is more apparent to readers if you present your analysis and results in the standard manner (as described above).

[Response] Thank you so much for your comment. As per your suggestion, we have now modified table 3 by re-ordering the list of interventions from the least to the most effective intervention. We also provided the stepwise calculations of ICERs.

6. since the difference in QALY is very small between strategies, I think that presenting COVID19 cases prevented would be more interesting than presenting QALYs, especially in a population less likely to die from COVID19. Most often in cost effectiveness analyses, both cases prevented and QALYs (gained) are presented as outcomes.

[Response] Thank you. As per your suggestion, we have added the number of infections for each intervention in Table 3. We also reported incremental cost-effectiveness per cases averted in the online appendix. 

Here are the instances that I interpreted as errors:

7. In the first paragraph of the results, you state that prevalence of 0.1% results in 351 cases. However, 351/20,500 = 1.7%. Likewise for 230/20,500 = 1.1% not 0.1%. and 1664/20,500=8.1% and 3126/20,500=15.2%.

[Response] Thank you. Sorry that we did a poor job of describing the estimation of prevalence. The number of cases among university affiliates is computed by estimating the transmission from members of the community off campus to those who are on campus as well as transmission between university affiliates while on campus. We do this because students are making contacts outside and inside the campus. The 351 cases we report reflect the estimated cumulative cases over the semester arising from transmission to university affiliates. Thus, at a 0.1% prevalence in the community surrounding the campus, when CDC guidelines are implemented, we expect to see 351 infections over the entire semester among university affiliates. (Our model did a very good job of predicting the actual number of infections at the Columbia campus over the course of the Fall 2020 semester.) We have made revisions throughout the document for clarity.

8. on p. 7, you state that you used 7 days for infectiousness. Table 2 lists exposure, symptoms, and infectiousness, with infectiousness of 9 days after symptom start. CDC lists the incubation period as 2-14 days, and infectiousness of 15 total days capturing 95% of all infectiousness. So, if 7 days was used, that is too low and doesn't match your table 2. You should consider using the standard 14 or 15 days of infectiousness.

[Response] Thank you. The “7” in the text was a typo. We corrected this typo in the revised paper.

As per your suggestion, we updated the days of infectiousness after symptoms onset in the revised model. Based on data from CDC (https://www.cdc.gov/coronavirus/2019-ncov/hcp/duration-isolation.html#:~:text=For%20most%20adults%20with%20COVID,with%20improvement%20of%20other%20symptoms.) and (https://www.journalofinfection.com/article/S0163-4453(20)30651-4/fulltext), it seems that for the majority of patients, the duration of infectiousness after symptoms onset is 10 days. Therefore, we updated our estimate for days of infectiousness after symptoms onset to 10 days (previously, it was 9 days). We also model these parameters probabilistically, so that in every Monte Carlo simulation run, a random number is generated from our distributions, representing individuals falling within tails of these distributions. 

9. it appears that your base case used 2-3 contacts per student. While you did conduct a sensitivity analysis in Figure 1 on this, with up to 10 contacts per student, I would have used 10 contacts per student as the base case. In tuberculosis studies, TB cases average 15 airborne contacts per case at diagnosis. I would think that COVID19 would result in a similar number of contacts, and possibly more for gregarious students.

[Response] Thank you for this observation. As suggested, we updated the number of daily contacts per student to 10 in the revised model. We also modified the statistical distribution for this parameter to gamma distribution based on a comment from another reviewer. We agree that many students, particularly gregarious students, or those who live with their family, will have more close contacts. At Columbia, the average number of close contacts per student reported in extended contact tracing in the Fall of 2020 was 2, but this number is likely an underestimate. 

Thanks again for your time and effort in helping us improve the paper!

6. PLOS authors have the option to publish the peer review history of their article (what does this mean?). If published, this will include your full peer review and any attached files.

Do you want your identity to be public for this peer review? For information about this choice, including consent withdrawal, please see our Privacy Policy.

Reviewer #1: No

Reviewer #2: Yes: Suzanne Marks

---

## [Decision Letter · Decision Letter 1]

21 Jul 2021

PONE-D-21-07113R1

The cost-effectiveness of common strategies for the prevention of transmission of SARS-CoV-2 in universities

PLOS ONE

Dear Dr. Zafari,

Thank you for submitting your manuscript to PLOS ONE. After careful consideration, we feel that it has merit but does not fully meet PLOS ONE’s publication criteria as it currently stands. Therefore, we invite you to submit a revised version of the manuscript that addresses the points raised during the review process.

We look forward to receiving your revised manuscript.

Kind regards,

Kevin Schwartzman

Academic Editor

PLOS ONE

Journal Requirements:

Additional Editor Comments (if provided):

Thank you for all your updates in response to the reviewer comments.

Reviewers' comments:

Reviewer's Responses to Questions

**Comments to the Author**

1. If the authors have adequately addressed your comments raised in a previous round of review and you feel that this manuscript is now acceptable for publication, you may indicate that here to bypass the “Comments to the Author” section, enter your conflict of interest statement in the “Confidential to Editor” section, and submit your "Accept" recommendation.

Reviewer #1: All comments have been addressed

2. Is the manuscript technically sound, and do the data support the conclusions?

Reviewer #1: Yes

3. Has the statistical analysis been performed appropriately and rigorously? 

Reviewer #1: Yes

4. Have the authors made all data underlying the findings in their manuscript fully available?

Reviewer #1: Yes

5. Is the manuscript presented in an intelligible fashion and written in standard English?

Reviewer #1: Yes

6. Review Comments to the Author

Reviewer #1: Thanks to the authors for addressing my comments. Only a few minor comments remain:

-Repeat word of "iterations" in the first sentence of Analyses

-Repeat word "actively" in the third sentence of Analyses

-In Table 3 - how should readers interpret negative ICERs? I suggest a footnote to explain this or to remove confidence intervals around ICERs since confidence intervals require context for interpretation (e.g., you can have a negative ICER in quadrant II and IV, and a positive ICER in quadrant I and III -- though interpretation of positive and negative ICERs vary GREATLY depending on quadrant). In the supplement, it appears most results fall in quadrant I and II. Please provide more clarity on interpretation in a footnote or text.

7. PLOS authors have the option to publish the peer review history of their article (what does this mean?). If published, this will include your full peer review and any attached files.

Reviewer #1: No

---

## [Author Response · Author response to Decision Letter 1]

8 Sep 2021

Reviewer #1: Thanks to the authors for addressing my comments. Only a few minor comments remain:

[Response] We would like to thank the reviewer for his/her excellent comments and help throughout the revisions process. Your comments throughout this process have greatly improved our paper. Thank you again for your time and invaluable comments/feedback.

-Repeat word of "iterations" in the first sentence of Analyses

[Response] Thank you. We have corrected this problem, which appears to be an error in PDF conversion. 

-Repeat word "actively" in the third sentence of Analyses

[Response] Thank you. As above, this appeared to be a conversion issue.

-In Table 3 - how should readers interpret negative ICERs? I suggest a footnote to explain this or to remove confidence intervals around ICERs since confidence intervals require context for interpretation (e.g., you can have a negative ICER in quadrant II and IV, and a positive ICER in quadrant I and III -- though interpretation of positive and negative ICERs vary GREATLY depending on quadrant). In the supplement, it appears most results fall in quadrant I and II. Please provide more clarity on interpretation in a footnote or text.

[Response] Thank you. As suggested, we removed the intervals around the ICERs in Table 3 and added a footnote that describes the meaning of negative ICERs in this table.

---

## [Editor Report · Decision Letter 2]

13 Sep 2021

The cost-effectiveness of common strategies for the prevention of transmission of SARS-CoV-2 in universities

PONE-D-21-07113R2

Dear Dr. Zafari,

We’re pleased to inform you that your manuscript has been judged scientifically suitable for publication and will be formally accepted for publication once it meets all outstanding technical requirements.

Kind regards,

Kevin Schwartzman

Academic Editor

PLOS ONE

Additional Editor Comments (optional):

Thank you for these final revisions.
---

## [Editor Report · Acceptance letter]

22 Sep 2021

PONE-D-21-07113R2 

The cost-effectiveness of common strategies for the prevention of transmission of SARS-CoV-2 in universities 

Dear Dr. Zafari:

I'm pleased to inform you that your manuscript has been deemed suitable for publication in PLOS ONE. Congratulations! Your manuscript is now with our production department. 

Kind regards, 

on behalf of

Dr. Kevin Schwartzman 

Academic Editor

PLOS ONE